# Comprehensive Analysis of Grain Production Based on Three-Stage Super-SBM DEA and Machine Learning in Hexi Corridor, China

Zhengxiao Yan [1], Wei Zhou [2,*], Yuyi Wang [3,4] and Xi Chen [3,4]

[1] Department of Civil and Environmental Engineering, FAMU-FSU College of Engineering, Florida State University, Tallahassee, FL 32310, USA; zy21@fsu.edu

[2] College of Agriculture and Food Science, Florida Agricultural and Mechanical University, Tallahassee, FL 32307, USA

[3] Institute of Surface-Earth System Science, School of Earth System Science, Tianjin University, Weijin Road 92, Tianjin 300072, China; yuyi_wang23@tju.edu.cn (Y.W.); xi_chen@tju.edu.cn (X.C.)

[4] Tianjin Key Laboratory of Earth Critical Zone Science and Sustainable Development in Bohai Rim, Tianjin University, Weijin Road 92, Tianjin 300072, China

* Correspondence: wei.zhou@famu.edu

**Abstract:** Food security is always a pressing agenda worldwide. The grain production in many areas has decreased due to the reduction in agricultural research funding and infrastructure investment. In this paper, we employed the Extreme-Tree algorithm to determine the main effectors in grain production in Hexi Corridor, Gansu, China, during 2002–2018. First, we applied the three-stage super-SBM DEA to precisely assess agricultural production. Then, we used the Extremely randomized trees algorithm to quantify the importance of each factor. Our results show that the variant of average efficiency score at the first stage was minimal. After removing the influence of environmental factors on production efficiency, the more accurate efficiency score was decreasing from 2002 to 2018. The $R^2$ value of the Extra-Tree model was 0.989 in the grain production analysis. Our research shows that grain production in the Hexi Corridor was controlled by human-driven but not nature-driven during our research period. Based on the importance attribution analysis of each model, it showed that the importance of human-driven investment occupied 93.7% of grain production. The importance of nature-driving was about 6.3%. Accordingly, we proposed corresponding opinions and suggestions to government and growers.

**Keywords:** grain production; three-stage super-SBM DEA; machine learning; Hexi Corridor

## 1. Introduction

Due to the aggravation of water shortage and the reduction in agricultural funds and infrastructure investment, the grain yield in many world areas has decreased or slowed down in recent years [1–3]. At the same time, it has been observed that visible climate change has impacted food security by increasing temperatures, changing precipitation patterns, and some more frequent extreme weather events [4]. Only if professionals guide farmers to accommodate climate change can they have the opportunity to make food supply not hindered by the adverse effects of climate change [5]. Although food production has grown exponentially in the past half-century, global food security is still facing great challenges under the influence of climate change [6–9], reduction in arable land [10], water shortage [11], urbanization [12], pandemics [13], and political factors [14]. About 8.9% of the world's people are still experiencing malnutrition to various degrees [15]. Therefore, solving the food security problem is still one of the most challenging and complicated problems globally.

The key to ensuring food security is closely associated with comprehensive grain production capacity growth. The main factors influencing grain production include the

input of grain production, grain production efficiency (GPE), and natural factors [16]. Efficiency plays a significant role in the growth of grain production, which is widely recognized by researchers and policymakers. Thiam et al. [17] emphasized the importance of efficiency to promote productivity, which has led to a surge in research on agricultural production efficiency all over the world. As the most populous country worldwide, China faces a severe problem of food scarcity that seriously affects the sustainable development of society [18]. The cultivated land area shifted from the wet south to the relatively dry north, leading to the continued decline of China's food self-sufficiency rate [19–21]. Therefore, how to improve the efficiency of grain production is one of the prominent problems faced by China and other developing countries in the sustainable development of society and economy.

Production efficiency (input-oriented) or technical efficiency (output-oriented), collectively called efficiency in this paper, is the performance of a system or a unit in the output of resource production [22]. According to the established factor input ratio, Farrell [23] believes that production efficiency refers to the ratio of the minimum cost required to produce a certain number of products to the actual cost under constant production technology and market price. Previous reports have applied various methods to explore and calculate how to improve agricultural grain production and its efficiency and proposed several policies. Alston et al. [24] reviewed agricultural development experience in developed countries and highlighted that increasing investment in agricultural R&D (research and development) will help improve global agricultural production efficiency. Ma et al. [25] indicated that China's agricultural productivity had tremendous spatial heterogeneity. It was reported that climate change significantly impacted agriculture, especially grain production and its efficiency. Warming had little effect on grain production but positively affected GPE [16]. The analysis of GPE can help reveal the advantages of the Decision-Making Unit (DMU) research and assist government and stakeholders in putting forward solutions and formulating policies to avoid the disadvantages. It should be noted that the premise of making sound recommendations and policies is an accurate assessment of the efficiency of each DMU.

SFA (Stochastic Frontier Analysis) and DEA (Data Envelopment Analysis) are two research methods of the Frontier Production Function (FPF). FPF constructs the functional relationship between input and output according to the given technical conditions and production factors. SFA is a parametric method to calculate the efficiency by constructing a specific formula including the production function or cost function. One of the disadvantages of SFA is that it is assumed there is no efficiency loss in all production units after excluding the influence of random factors. The research results equate the potential production capacity with the actual production capacity, which not only affects the accuracy of prediction but also makes it difficult to judge the potential production increase space. Additionally, SFA is mainly analyzed by constructing functions, so its specific form is challenging to optimize and update; at this point, DEA has excellent advantages. DEA is by creating a minimum possibility set to determine the relatively effective efficiency frontier and scores of the efficiency of each by mathematical programming [26]. So far, DEA has been developed for many more accurate models, such as dynamic network DEA, super-efficiency DEA, SBM-DEA, etc. Most importantly, SFA is ineffective when there is a high correlation between input and output data, while DEA does not have this concern [27,28]. The input-output data of this study (see Section 2 for data description) are relatively highly correlated (Figure 1). Therefore, it is more reasonable to use DEA instead of SFA theoretically to evaluate the efficiency in this paper.

Originally, DEA was proposed as a nonparametric model to evaluate the relative efficiency of a group of comparable DMUs containing multiple input and output variables [29]. DEA can avoid artificial subjectivity because the weight of each index is automatically calculated according to the specific data of input and output. However, the traditional DEA cannot sort the effective DMU, so Andersen and Petersen [30] proposed a super-efficiency model so that the DMU can also compare the efficiency when it reaches the

frontier. Yang et al. [31] applied the super-efficiency DEA to calculate the environmental efficiency of 30 provinces in China. The efficiency in China's relatively poor and arid northwest was significantly lower than that in the prosperous and humid southeast. Shuai and Fan [32] measured the efficiency of China's green economy through super-efficiency DEA. The results showed that the impact of environmental regulation on efficiency first decreased and then increased. Nevertheless, the input and output of traditional DEA are enlarged or reduced in the same proportion. When there are slacks in input or output, it is easy to overestimate the efficiency of DMUs and affect the accuracy of the results. Therefore, Tone [33] proposed Slack Based Model-DEA (SBM-DEA) to address the slack issue and the proportional change in inputs or outputs. The radial DEA and SBM-DEA were applied to calculate the efficiency of Polish winter wheat. It is shown that the efficiency scores generated by SBM-DEA are lower, which can better distinguish the efficiency difference of DMUs producing winter wheat [34]. Tone [35] proposed super-SBM DEA, which combines the super-efficiency model with the non-radial model. It can evaluate the efficiency of the effective or ineffective DMUs more accurately. Zhang et al. [36] constructed super-SBM DEA to measure China's low carbon economy efficiency. It is said that the model can effectively rank provinces with different efficiency. There are other applications of super-SBM DEA in various aspects, which prove that this model can effectively improve the shortcomings of traditional DEA [37–41]. However, the applications in agriculture are still relatively few [42].

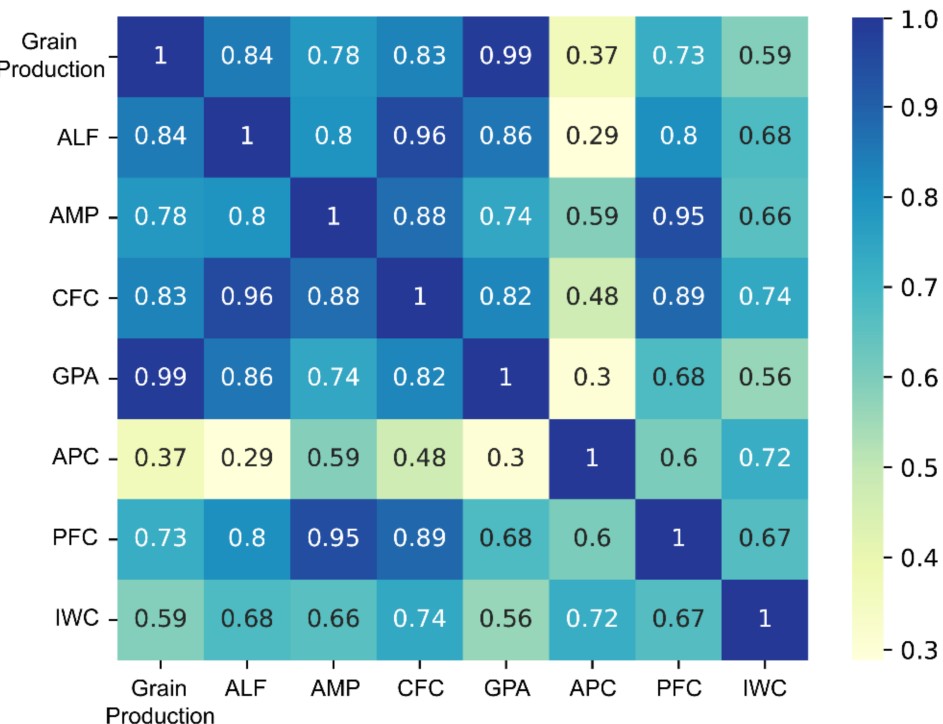

**Figure 1.** Correlation matrix between the output and input.

The traditional DEA has a severe deficiency compared with SFA. Given only the controllable factors of DMUs, it is difficult to judge whether low efficiency is due to managerial inefficiencies, external environmental factors, or stochastic errors. Many scholars have realized that this problem cannot be ignored and began to explore its solutions, put forward some ideas and theories, and paved the way for the follow-up research [43,44]. Fried et al. [45,46] discussed incorporating external environmental factors and random errors into DEA models to remove their effect on actual efficiency. A three-stage DEA based on the combination of traditional DEA and SFA methods is proposed [46]. DEA is used to evaluate the efficiency of DMUs and SFA to eliminate the influence of external environmental factors and stochastic errors on the results. Hence, this method has been

applied in many fields to assess efficiency, for instance, measuring agricultural production efficiency [47], environmental and eco-efficiency [48], banking systems efficiency [49], etc. However, the combination of three-stage super-SBM DEA is rarely used, especially for measuring GPE.

The quantitative study of GPE is crucial for the comprehensive analysis of grain production. With the gradual deepening of the research on grain production worldwide, scholars and government departments have realized that it is necessary to carry out a comprehensive evaluation to systematically understand grain production's current situation and provide the scientific basis for administrative departments. Climate change and its impact assessment have become a hot topic recently worldwide. Although it seems that the rise in $CO_2$ concentration may have a positive effect on grain production, climate change will make grain production more unstable because of the increasing temperature, the frequency of extreme rainstorms, and droughts [16,50]. Therefore, we conduct a quantitative study on the impact of various input factors, grain production efficiency, and natural factors on grain production. Currently, the attribution of factors affecting grain production is mainly studied by constructing the elastic coefficient of the production function [50–53]. In addition, there are many other methods for regression analysis of grain production factors, such as the multiple linear regression model [51], the spatial regression models [52], the grey forecasting model [53], and so on.

In recent years, the development of artificial intelligence has entered a new stage, showing unique characteristics, such as the integration of AI technology and other disciplines, and showing a rapid development trend in many fields [54]. This has dramatically fostered applications in geosciences and provided a new way to solve all kinds of geoscientific problems [55–58]. It is a new exploration and attempts to apply machine learning in the analysis of the influencing factors of agricultural grain production. Random Forest can deal with nonlinear data, which is suitable for high-dimensional data. The importance of each feature can be obtained directly from the training results. The Extra-Trees is a variant algorithm of the Random Forest [59]. Random Forest and Extra-Trees are similar in the algorithm. The difference is that the Random Forest uses the bagging model, but the Extra-Trees randomly selected features of all samples. Therefore, the goodness of fit of Extra-Trees is better than Random Forest to some extent. Aparecido et al. [60] believed that the Extra-Trees regression model performs better than others in predicting cotton yield after adding climate data. Since this paper studies agricultural production also, we applied the Extra-Trees algorithm of integrated learning to analyze the influencing factors of grain production and then determine the importance of each factor.

This paper's main innovations and research contents are as follows: (1) Most of the measurements of grain production efficiency take large-scale regions (such as countries, provinces/states) as DMUs. However, our research takes five cities in the Hexi Corridor of Gansu Province as DMUs to explore the more accurate temporal and spatial variation of grain production efficiency on small and medium-sized scales; (2) By combining the advantages of three-stage DEA, Super-DEA, and SBM-DEA, our research attempts to explore and establish a perfect combination model of three-stage super-SBM DEA to precisely assess agricultural production; (3) Our research aims to employ the Extreme-Tree algorithm to analyze the influencing factors of grain production, including the input of various production factors, grain production efficiency, and natural factors, to determine whether humans mainly drive the grain production or nature.

## 2. Materials and Methods

### 2.1. Study Area

The Hexi Corridor is located in the arid region of Northwest China between 92°12′ E–104°20′ E and 37°17′ N–42°48′ N, with a total area of 270,000 km$^2$ (Figure 2). Typically, the Hexi Corridor refers to Jiuquan, Jiayuguan, Zhangye, Jinchang, and Wuwei. The Hexi Corridor belongs to the arid continental climate. Although there is little precipitation (only about 200 mm annually), other climatic conditions for the development of agriculture

are still superior. For example, the annual duration of sunshine in this area can reach 2550–3500 h, which is very beneficial to the growth of crops. Additionally, Qilian Mountain is rich in ice and snow meltwater, so irrigation agriculture is developed there. Moreover, due to the rapid urbanization in southeast coastal areas of China, the focus on grain production has shifted to the northwest. Therefore, Hexi Corridor is the most important commercial grain base in Northwest China, and we chose it as the study area in this research. China's national map is from the Standard Map Service System (http://bzdt. ch.mnr.gov.cn/, last accessed on 25 July 2021), and the DEM of Gansu Province is from the Resource and Environment Science and Data Center (https://www.resdc.cn/, last accessed on 25 July 2021).

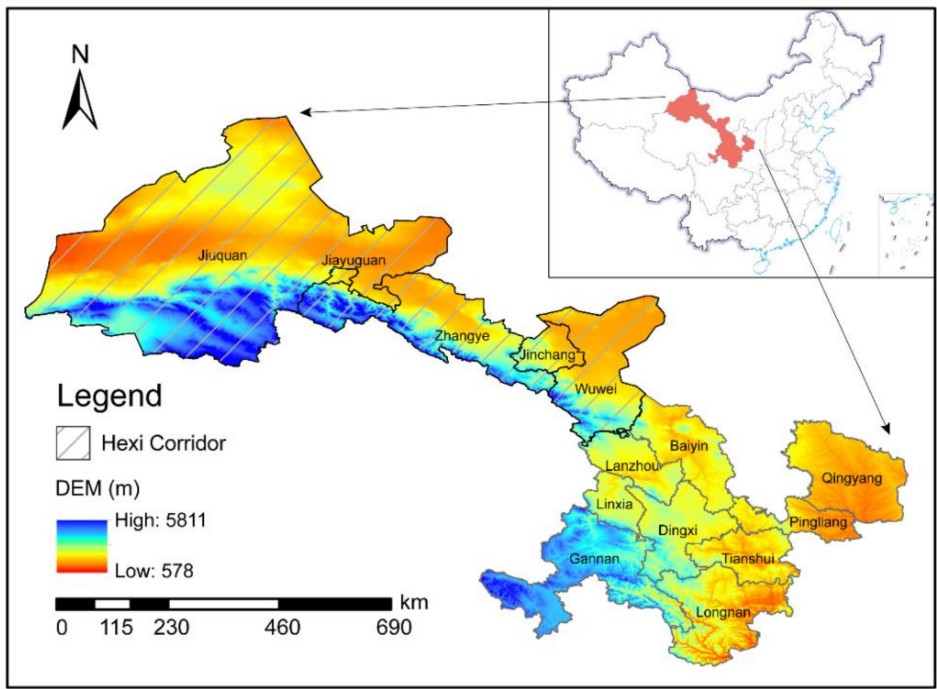

**Figure 2.** Location and digital elevation model (DEM) of Hexi Corridor and Gansu Province.

*2.2. Methodology*

2.2.1. Three-Stage Super-SBM DEA

As discussed in the introduction, the DEA model is a nonparametric method to evaluate efficiency, which can deal with the high correlation between variables. DEA model calculates the weight of each variable according to the data, which can avoid human influence. The formula of the traditional input-oriented DEA model is as follows:

$$
\begin{aligned}
&min\theta \\
&s.t. \begin{cases}
\sum\limits_{j=1}^{n} x_{ij}\lambda_j \leq \theta x_{ik} \\
\sum\limits_{j=1}^{n} y_{rj}\lambda_j \geq y_{rk} \\
\lambda_j > 0, \ \sum\limits_{j=1}^{n} \lambda_j = 1 \\
i = 1, 2, \ldots, m; \ r = 1, 2, \ldots, q; \ j = 1, 2, \ldots, n
\end{cases}
\end{aligned} \tag{1}
$$

where $\theta$ is the efficiency score of DEA, which is in the range of [0, 1]; $\lambda$ is the linear combination coefficient of each DMU; $x_i$ denotes the input of each DMU; $y_{rj}$ denotes the output of each DMU.

Fried [46] claimed that the efficiency scores directly calculated by DEA are influenced by three main aspects: managerial inefficiencies, external environmental factors, and

stochastic errors. However, the traditional DEA does not consider removing external environmental factors and stochastic errors, which will have a certain impact on the results of DMU efficiency evaluation. This three-stage process can effectively remove the impacts of external environmental factors and stochastic errors on the efficiency scores so that we can obtain the accurate efficiency scores that are only influenced by mismanagement to carry out follow-up research better [47,48,61–63]. The three-stage process can remove the influence of external environmental factors and stochastic errors; The super-DEA can sort all effective DMUs that have reached the efficiency frontier; The SBM-DEA can make the non-effective DMUs not have to follow the radial direction to change in the same proportion and maximize the degree of DMUs prediction accuracy that does not reach the efficiency frontier. Thus, we selected the three-stage super-SBM DEA to calculate the GPE in this research. Each step is illustrated in detail in the following stages:

Stage 1: The Super-SBM DEA

In our research, a super-SBM DEA is applied to assess the grain production efficiency (GPE). Super-SBM DEA is a combination of SBM-DEA and super-DEA [33]. It overcomes the issue that the scores of traditional DEA cannot be compared when they reach the efficiency frontier, and can deal with the overestimation of non-effective DMUs (Decision-Making Units) by radial DEA [35–38,64]. Following the Tone [35] model, we constructed a production system with n DMUs and each of them includes input factors and output factors. Input factors can be denoted by vectors $x \in R^p$ and output factors can be denoted by vectors $y \in R^q$, where $p$ and $q$ are the number of inputs and outputs, respectively. So, the corresponding input matrix is set to $X = [x_1, x_2, \ldots, x_n] \in R^{p \times n}$, and the output matrix is set to $Y = [y_1, y_2, \ldots, y_n] \in R^{q \times n}$. The production possibility is as follows:

$$p(x) = \{(x, y) | x \geq X\lambda, \ y \geq Y\lambda, \ \lambda \geq 0\} \tag{2}$$

where the $\lambda$ is a non-negative vector of weights allocated to the inputs and outputs. The formula of the SBM-DEA is as follows [33]:

$$min\rho = 1 - \frac{1}{p} \sum_{i=1}^{p} \frac{q_i^-}{x_{i0}} \Big/ 1 + \frac{1}{q} \sum_{r=1}^{q} \frac{q_r}{y_{r0}}$$

$$s.t. \begin{cases} x_{i0} = \sum_{j=1}^{n} x_{ij}\lambda_j + q_i^- \\ y_{r0} = \sum_{j=1}^{n} y_{rj}\lambda_j - q_r \\ q_i^-, q_r, \lambda_j > 0, \ \sum \lambda_j = 1 \\ i = 1, 2, \ldots, p; \ r = 1, 2, \ldots, q \end{cases} \tag{3}$$

where $\rho$ is the efficiency score of SBM-DEA, which is in the range of [0, 1]; $q^-$ is the slack vector of the excess input; $q$ is the slack vector of the shortage output. The subscript 0 represents the estimated DMU. The DMU is inefficient when $\rho < 1$. Only when $\rho = 1$ and $q^- = q = 0$ is the DMU is efficient. However, when multiple DMUs are efficient simultaneously, we need to further apply the super-SBM DEA to sort the DMUs that reach the frontier. The super-SBM DEA is as follows [35]:

$$min\delta = \frac{1}{p} \sum_{i=1}^{p} \frac{\overline{x}}{x_{i0}} \Big/ \frac{1}{q} \sum_{r=1}^{q} \frac{\overline{y}}{y_{r0}}$$

$$s.t. \begin{cases} \overline{x} \geq \sum_{j=1, \neq 0}^{n} x_{ij}\lambda_j \\ \overline{y} \leq \sum_{j=1, \neq 0}^{n} y_{rj}\lambda_j \\ \overline{x} \geq x_{ij}, \ \overline{y} \leq y_{rj}, \ \lambda_j > 0 \\ i = 1, 2, \ldots, p; \ r = 1, 2, \ldots, q \end{cases} \tag{4}$$

where $\delta$ is the efficiency score of super-SBM DEA which can be over 1. The other same variables have the same definitions in Equation (3). The $\overline{x}$ is the mean vector of inputs vector, and the $\overline{y}$ is the mean vector of outputs vector. The super-SBM DEA is able to obtain a more precise efficiency result with the consideration of slacks, inefficient and efficient DMUs. To solve the problem of the slacks of variables and the measurement error caused by the radial direction, the non-radial and non-oriented SBM-DEM based on the slacks can measure the efficiency more strictly and accurately with significant advantages. This study first used the SBM-DEM model for preliminary assessment but found that many cities in the Hexi corridor have reached the efficiency frontier in different years, so it is impossible to rank each city specifically. Thus, the super-SBM DEA model considering super efficiency is adopted to assess the grain production efficiency further and rank the efficiency of each DMU in this study.

Stage 2: The Stochastic Frontier Analysis (SFA)

To improve and obtain better results from DEA only influenced by mismanagement, we applied the SFA to eliminate the impacts of external environmental factors and stochastic errors on the efficiency scores [46]. The introduction of slacks in linear programming is to transform inequality constraints into better operational equations. The slacks in DEA are the difference or the movement in original and projection of each input and output in DMUs, which can represent the inefficiency in scores. Thus, we built an SFA regression model, in which the slacks are as the dependent variable; managerial inefficiencies, the external environmental factors, and stochastic errors are as independent variables, as in Equation (5):

$$S_{ni} = f(Z_i; \beta^n) + v_{ni} + \mu_{ni}; \; i = 1, 2, \ldots, I; \; n = 1, 2, \ldots, N \tag{5}$$

where $S_{ni}$ is the slack of $n$-th input of $i$-th DMU; $Z_i$ is the vector of external environmental factors likely to affect the efficiency; $\beta^n$ is the coefficient of external environmental factors; $f(Z_i; \beta^n)$ represents the effects of external environmental factors on the slacks; $v_{ni}$ is the stochastic error; $\mu_{ni}$ is the managerial inefficiency; $v_{ni} + \mu_{ni}$ indicates the mixed error term. It should be noted that normally $v \sim N(0, \sigma_v{}^2)$ and $\mu \sim N^+(0, \sigma_\mu{}^2)$. We can directly obtain the external environmental factors part using the SFA regression model and then use Equation (5) to get the mixed errors. However, it is hard to separate the stochastic errors, which we actually need, from mixed errors. As a result, we applied Equation (6) to calculate the section of managerial inefficiency in total mixed errors [47]:

$$E(\mu|\varepsilon) = \sigma_* \left[ \frac{\varphi\left(\frac{\lambda\varepsilon}{\sigma}\right)}{\phi\left(\frac{\lambda\varepsilon}{\sigma}\right)} + \frac{\lambda\varepsilon}{\sigma} \right], \; \sigma_* = \frac{\sigma_\mu \sigma_v}{\sigma}, \sigma = \sqrt{\sigma_\mu{}^2 + \sigma_v{}^2}, \; \lambda = \sigma_\mu / \sigma_v \tag{6}$$

where $\sigma_\mu$ is the standard deviation (SD) of managerial inefficiencies; $\sigma_v$ is the SD of stochastic errors; $\varphi$ is the probability density function (PDF) of the standard normal distribution; $\phi$ the cumulative probability function (CDF) of the standard normal distribution; $\varepsilon$ is the mixed error; $E(\mu|\varepsilon)$ is the managerial inefficiency. Then, we could calculate the stochastic errors by subtracting managerial inefficiencies from mixed errors. Eventually, we employed Equation (7) to put all DMUs under the same environmental and nonrandom conditions, so as to obtain the adjusted input only affected by mismanagement:

$$x^*_{ni} = x_{ni} + [\max(f(Z_i; \beta^n)) - f(Z_i; \beta^n)] + [\max(v_{ni}) - v_{ni}]; i = 1, 2, \cdots, I; n = 1, 2, \cdots, N \tag{7}$$

where $x^*_{ni}$ is the input after adjustment; $x_{ni}$ is the input before adjustment; $[\max(f(Z_i; \beta^n)) - f(Z_i; \beta^n)]$ is to adjust the external environmental factors to the same level; $[\max(v_{ni}) - v_{ni}]$ is to adjust the stochastic errors to the same level.

Stage 3: the adjusted DEA

Stage 3 is the last step. The adjusted inputs (usually smaller than before because the impact of external environmental factors and stochastic errors is removed) are very

different from those before the adjustment. The adjusted input data were used to re-assess the GPE by super-SBM DEA, which can precisely demonstrate the better efficiency scores without the influence of external environmental factors and stochastic errors with only mismanagement.

### 2.2.2. Extra-Trees Algorithm

Machine learning is very good at addressing nonlinear and multi-variable regression problems. It is difficult to determine how the factors intricately affect grain production through linear regression. To tackle this problem, we applied the Extra-Trees (Extremely randomized trees) algorithm to quantify the importance of each factor [59]. Extra-Trees is a tree-based ensemble supervised machine learning algorithm, and it has been broadly used in regression and classification tasks in many fields [60]. In this research, we randomly set 70% of the dataset for training and 30% for testing using the scikit-learn library in Python [61]. We then utilized the unadjusted $R^2$ without correcting for bias in order to evaluate the performance of this algorithm, as follows:

$$R^2(y, \hat{y}) = 1 - \frac{\sum\limits_{i=1}^{n} (y_i - \hat{y}_i)^2}{\sum\limits_{i=1}^{n} (y_i - \overline{y})^2} \tag{8}$$

where $\overline{y} = \frac{1}{n} \sum\limits_{i=1}^{n} y_i$. The larger the $R^2(y, \hat{y})$, the better the simulation effect. We would use the predict function to calculate the $R^2(y, \hat{y})$ between the observed and the predicted to see the performance in the training set, calibration set, and total.

### 2.3. Data Sources, Descriptions, and Preprocessing

This study contains three main categories of variables for super-SBM DEA, stage-2 SFA, and Extra-Trees. The variables were selected by practicality, availability, and previous relevant studies.

### 2.3.1. Variables for Super-SBM DEA

The variables for super-SBM DEA are composed of inputs and the output. Agricultural Labor Force, Total Agricultural Machinery Power, Chemical Fertilizer Consumption, and Grain Crops Planting Area were taken from the Gansu Development Yearbook (2003–2019, https://data.cnki.net/yearbook/Single/N2021020070, last accessed on 31 July 2021). Agricultural Pesticide Consumption and Agricultural Plastic Film Consumption were collected from Gansu Rural Yearbook (2003–2019, https://data.cnki.net/Yearbook/Single/N2021040103, last accessed on 31 July 2021). Grain Production and Irrigation Water Consumption were obtained from Gansu Water Resources Bulletin (2003–2019, http://slt.gansu.gov.cn/slt/c106726/c106732/c106773/zcfg.shtml, last accessed on 31 July 2021). Note that the Yearbooks are basically one year later than the actual data because they were finished at the time of publication.

According to the newly revised China Agricultural Mechanization Management Statistical Investigation System (http://www.njhs.moa.gov.cn/tzggjzcjd/201902/t20190220_6314910.htm, last accessed on 31 July 2021), agricultural transport vehicles, three-wheeled vehicles, and low-speed trucks were deleted from the statistical indicators of Total Agricultural Machinery Power. The data of Total Power of Agricultural Machinery after 2016 have changed a lot in line with the previous data. Consequently, we collected the most important statistical factors affecting the Total Agricultural Machinery Power ($y$) that we could obtain, including the number of large and medium-sized tractors ($x_1$), small tractors ($x_2$), combined harvesters ($x_3$), and motorized threshers ($x_4$). We then used the multiple linear regression model to fit the $\ln(y)$ between $\ln(x_1)$, $\ln(x_2)$, $\ln(x_3)$, and $\ln(x_4)$. This regression rejected the null hypothesis and passed the significance test ($Prob > F = 0.0000$; $R^2 = 0.9895$).

Lastly, we applied this regression model to predict the Total Agricultural Machinery Power of each DMU each year.

### 2.3.2. Variables for Stage-2 SFA

The variables for stage-2 SFA are environmental variables to eliminate the effects of external environmental factors. The selection principle of environmental variables is that they impact DMUs, but DMUs do not control them and can stand for the environmental level. The environmental variables included Per Capita Gross Regional Product (GRP), standing for Economic development level, Proportion of Primary Industry in GRP, standing for Industrial structure, Proportion of Inner Expenditures of Scientific and Technological (S&T) Activity in GRP, standing for Scientific innovation level, and Total Value of Imports and Exports Commodities standing for Degree of openness. These four kinds of data were collected from the Gansu Development Yearbook. More details can be found in Table 1 and below.

**Table 1.** Summary of all the variables in this paper.

| Variable | Category | Description | Unit |
|---|---|---|---|
| Agricultural Labor Force (ALF) | IV of DEA; IV of ETS (HD) | The number of people actually participating in agricultural labor | $10^4$ people |
| Total Agricultural Machinery Power (AMP) | IV of DEA; IV of ETS (HD) | The sum of the power of all agricultural machinery | kWh |
| Chemical Fertilizer Consumption (CFC) | IV of DEA; IV of ETS (HD) | The amount of chemical fertilizer actually used in agricultural production, converted to pure amount | ton |
| Grain Crops Planting Area (GPA) | IV of DEA; IV of ETS (HD) | The sown or transplanted area of grain crops harvested by agricultural producers on all land | hectare |
| Agricultural Pesticide Consumption (APC) | IV of DEA; IV of ETS (HD) | The amount of pesticide actually used in agricultural production | ton |
| Agricultural Plastic Film Consumption (PFC) | IV of DEA; IV of ETS (HD) | The amount of plastic film actually used in agricultural production | ton |
| Irrigation Water Consumption (IWC) | IV of DEA; IV of ETS (HD) | The amount of water introduced from the water source for irrigation in the area | $10^8$ $m^3$ |
| Grain Production | OV of DEA; OV of ETS | The total amount of grain produced by agricultural producers | ton |
| Per Capita Gross Regional Product | EV1 of SFA | The per capita final results of production activities of all resident units in the region | CNY |
| Proportion of Primary Industry in GRP | EV2 of SFA | Ratio of primary industry to GRP | % |
| Proportion of Inner Expenditures of S&T Activity in GRP | EV3 of SFA | Ratio of inner expenditures of S&T activity to GRP | % |
| Total Value of Imports and Exports Commodities | EV4 of SFA | The total amount of goods actually entering and leaving China | $10^4$ CNY |
| Grain Production Efficiency (GPE) | IV of ETS (HD) | The performance of a DMU in the output of grain production | Dimensionless |
| Annual Precipitation (AP) | IV of ETS (ND) | The amount of total precipitation depth in one year | mm |
| Annual Average Temperature (AAT) | IV of ETS (ND) | The mean of daily average temperature of each day in the whole year | °C |
| Average Sunshine Duration (ASD) | IV of ETS (ND) | An indicator measuring the daily duration of sunshine | hour |
| Area Covered by Natural Disaster (ACD) | IV of ETS (ND) | The sown area of crops reduced by more than 10% due to disasters | hectare |
| $CO_2$ Emissions ($CO_2$) | IV of ETS (HD) | The emissions stemming from the burning of fossil fuels and the manufacture of cement | $10^6$ tons |

Note: Input variable = IV; Output variable = OV; Environmental variable = EV; Extra-Trees = ETS.

Note that the Per Capita Gross Regional Product and the Total Value of Imports and Exports Commodities are affected by inflation, but when the official department counts, the statistics are the current prices. Therefore, we adjusted the data by GDP deflator (annual inflation rate) of China from the World Bank-set 2002 as the benchmark. Furthermore, the Inner Expenditures of S&T Activity were recorded in US dollar from 2002 to 2013. We converted the CNY into the US dollar by the official exchange rate from the World Bank (https://data.worldbank.org/, last accessed on 2 August 2021).

### 2.3.3. Variables for Extra-Trees

The variables for Extra-Trees contain all the input and output variables in assessing the GPE, the GPE itself, and natural factors that specifically include Annual Precipitation, Annual Average Temperature, Average Sunshine Duration, Area Covered by Natural Disaster, and $CO_2$ Emissions. We acquired Annual Precipitation and Area Covered by Natural Disaster from Gansu Water Resources Bulletin, Annual Average Temperature and Average Sunshine Duration from China Meteorological Science Data Sharing Service Network (http://data.cma.cn/, last accessed on 2 August 2021), and the $CO_2$ Emissions from Shan et al. [62,63]. We then would like to classify all input variables into two categories, human-driven (HD) and nature-driven (ND), to find out the importance of the impact of these two categories on grain production, respectively.

## 3. Results

### 3.1. Stage 1—Application of the Super-SBM DEA

We used super-SBM DEA to assess the GPE from 2002 to 2018 in the Hexi Corridor for the first stage. Figure 3 showed that the average trend was flat, and the mean was 0.877 throughout the period ignoring the effects of external environmental factors and stochastic errors. The GPE of most cities in the Hexi Corridor decreased from the start to the middle, around 2010, and then increased at the end of the research period. Although the GPE of Jiuquan decreased most obviously in the whole period, it still showed an upward trend after 2011. The GPE of these five cities was in descending order: Jiayuguan (1.017), Zhangye (0.991), Jinchang (0.949), Wuwei (0.882), and Jiuquan (0.546). However, stage 1 did not consider the external environmental factors and stochastic errors in the assessment of the GPE, which cannot reflect the real GPE. So, further steps of adjustment are necessary to be taken.

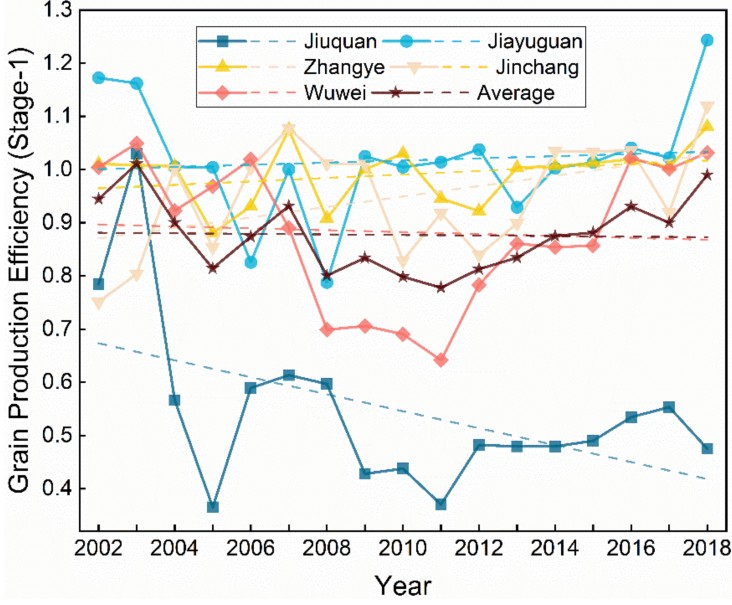

**Figure 3.** The GPE of Stage 1 in the Hexi Corridor from 2002 to 2018. Note: Dash lines were fitted by linear regression.

### 3.2. Stage 2—SFA Analysis Results

In the second stage, slacks of each input variable in each DMU from stage 1 were taken as dependent variables, and the Per Capita Gross Regional Product (EV1), Proportion of Primary Industry in GRP (EV2), Proportion of Inner Expenditures of S&T Activity in GRP (EV3), and Total Value of Imports and Exports Commodities (EV4) were chosen as independent variables for SFA. The LR tests of the one-sided error were vital for SFA. All seven SFA regressions passed the 1% significance test in this stage. Therefore, they rejected the null hypothesis, and it was sensible to use SFA in stage 2. $\gamma$ is the ratio of the variance of managerial inefficiencies to the variance of mixed errors. Therefore, it can reflect which one has a more significant impact on the mixed errors, management inefficiencies, or stochastic errors. It indicates that the mixing errors are mainly dominated by management inefficiencies when gamma approaches 1. In this study, each $\gamma$ was over 0.98, implying that the mismanagement dominated the inefficiencies (Table 2).

**Table 2.** SFA regression results of Stage 2.

| | ALF | AMP | CFC | GPA | APC | PFC | IWC |
|---|---|---|---|---|---|---|---|
| Constant | 1.202 *** | 54,709.136 *** | 8940.960 *** | −2644.35 *** | 383.137 * | 651.793 | −0.748 * |
| | (3.233) | (54,627.162) | (5684.217) | (−75.312) | (1.693) | (1.581) | (−1.884) |
| EV1 | 0.000 ** | 3.578 * | −0.176 ** | 0.031 ** | 0.006 | 0.005 | 0.000 |
| | (−2.225) | (1.966) | (−2.117) | (2.313) | (0.491) | (0.270) | (1.030) |
| EV2 | −6.197 *** | −1,177,984.300 *** | −40,455.221 *** | 5672.075 *** | −3172.5 *** | −4686.751 *** | 2.478 *** |
| | (−7.979) | (−1,177,631.100) | (−20,828.845) | (39.612) | (−4.674) | (−6.090) | (3.092) |
| EV3 | −131.930 *** | −13,917,628.00 *** | −553,435.43 *** | 4193.428 *** | −55,454.8 *** | −91,847.74 *** | −113.6 *** |
| | (−131.823) | (−13,917,609.000) | (−545,029.550) | (2900.085) | (−539.108) | (−863.222) | (−117.229) |
| EV4 | 0.000 *** | −0.136 | 0.000 | 0.001 | −0.001 * | −0.001 | 0.000 *** |
| | (10.645) | (−0.934) | (−0.018) | (0.654) | (−1.864) | (−0.940) | (27.155) |
| $\sigma^2$ | 111.142 | 615,589,410,000.00 | 1,322,312,200.00 | 19,830,247.00 | 9,316,170.90 | 25,419,789.00 | 43.471 |
| $\gamma$ | 1.000 | 0.982 | 0.994 | 1.000 | 0.994 | 0.995 | 1.000 |
| LR | 54.104 | 33.478 | 36.930 | 69.548 | 44.653 | 33.626 | 58.976 |

Note: Data in parentheses are t-statistics. *, **, *** represent the 10%, 5%, 1% significance, respectively.

At the time of pondering the influence of environmental variables on the input slacks, if the constant coefficient is positive in this case, the increase in environmental factors will cause a rise in the input slacks, denoting that the rise in environmental factors is not conducive to the GPE. On the contrary, if the constant coefficient is negative, the increase in environmental factors will lead to a drop in the input slacks, indicating that it is conducive to the GPE. The SFA regression results show that these four environmental variables influenced the slacks of input to some extent (Table 2), as follows:

(1)    Per Capita Gross Regional Product (GRP)

Per Capita GRP was negatively correlated with the slacks of the Chemical Fertilizer Consumption and positively correlated with the slacks of the Agricultural Labor Force, Total Agricultural Machinery Power, and Grain Crops Planting Area significantly. The promotion of Per Capita GRP would increase the slacks of positively correlated variables. The higher the Per Capita GRP, the more cutting-edge technology and machines would be applied in agriculture. It was likely to improve GPE to a certain extent. As a result, with the increase in GRP per capita, the slacks of ALF, AMP, and GPA would rise, but interestingly, the slack of CFC will become less. This indicated that more fertilizer might be needed to maintain or enhance GPE.

(2)    Proportion of Primary Industry in GRP

The coefficients of this indicator were statistically negative for ALF, AMP, CFC, APC, and PFC and positive for GPA and IWC. All the t-statistics had passed the 1% significance test. It indicated that the increase in the Proportion of Primary Industry in GRP would result in the drop of slacks of the ALF, AMP, CFC, and APC and the rise of slacks of the GPA and IWC. Most land and water resources will be provided to agriculture rather than urban residents, industry, or commerce when EV2 increases. It may cause waste of land and water resources, thus increasing slacks. However, generally, the increase in the Proportion

of Primary Industry in GRP would make most of the input slacks less and the efficiency scores better.

(3)　Proportion of Inner Expenditures of S&T Activity in GRP

All these t-statistics of the coefficients had passed the 1% significance test as well. There was a statistically negative correlation with all input slacks except for the Grain Crops Planting Area. This means that this environmental variable played a significant role in the input slacks like EV2. The increase in the Proportion of Inner Expenditures of S&T Activity in GRP would lead to the majority of the slacks going down and improving efficiency.

(4)　Total Value of Imports and Exports Commodities

This environmental variable was negatively correlated with the slacks of Agricultural Pesticide Consumption and positively correlated with the slacks of the Agricultural Labor Force and Irrigation Water Consumption. ALF and IWC passed the 1% significance test. The increase in Total Value of Imports and Exports Commodities will make the increase slacks of ALF and IWC, which means, in the probabilistic view, the rise in EV4 was more likely to decrease the GPE.

*3.3. Stage 3—Analysis of the GPE and Adjusted DEA*

The input data in stage 1 can be adjusted by putting all DMUs under the same circumstances using the Equations (4)–(6) and the results of SFA regression in order to obtain the new input data without the impacts of external environmental factors and stochastic errors. Figure 4 showed that the average trend slightly decreased instead of flattened in stage 1, and the mean value was 0.809 throughout the period, lower than 0.877. The GPE average trends of stage 3 in Jiuquan, Zhangye, Jinchang, and Wuwei were approximately the same as in stage 1. However, Jiayuguan had a dramatic change, from a slow rise to a rapid decline. Moreover, we can see that stage 3 had a significant reduction compared with stage 1 because the external environmental factors in stage 2 had a particular promoting effect for relatively prosperous cities on the efficiency scores, as we mentioned above. Therefore, the scores of stage 1 would be higher than those of stage 3. Additionally, the variance of the GPE in stage 3 was much more significant than that in stage 1, except for Wuwei. It proved that the external environmental factors had caused notable fluctuations in the variance of efficiency scores. Although the fluctuation was relatively apparent in the study period, Jiuquan, Zhangye, Jinchang, and Wuwei had shown a slow upward trend since around 2010 after falling or fluctuating. Jiayuguan was not applied in such a rule, and its average GPE decreased the most during the study period from stages 1 to 3. The GPE of these five cities was in descending order: Zhangye (0.942), Jinchang (0.821), Wuwei (0.812), Jiayuguan (0.755), and Jiuquan (0.713).

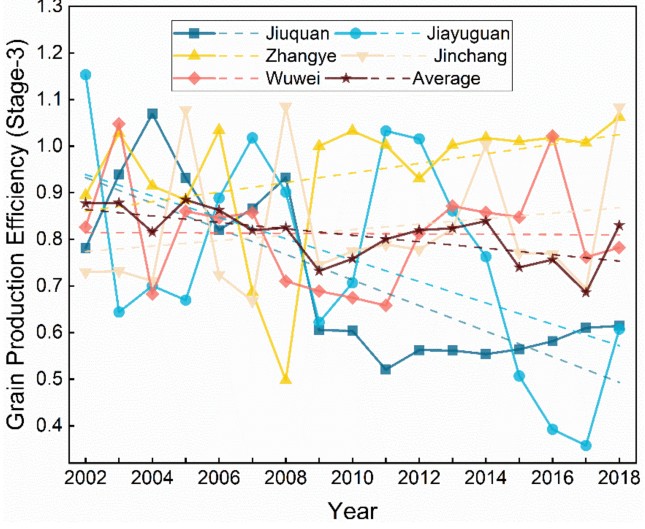

**Figure 4.** The GPE of Stage 3 in the Hexi Corridor from 2002 to 2018.

### 3.4. Spatial Characteristics of Grain Production Efficiency in the Hexi Corridor

We classified the grain production efficiency data of five administrative divisions of Hexi Corridor, Jiuquan, Jiayuguan, Zhagnye, Jinchang, and Wuwei. For the convenience of analysis, the average GPE of four-time sections of 2002–2005, 2006–2009, 2010–2013, and 2014–2018 were selected for study. The spatial distribution characteristics of the grain production efficiency pattern were figured out by ArcGIS platform (Figure 5).

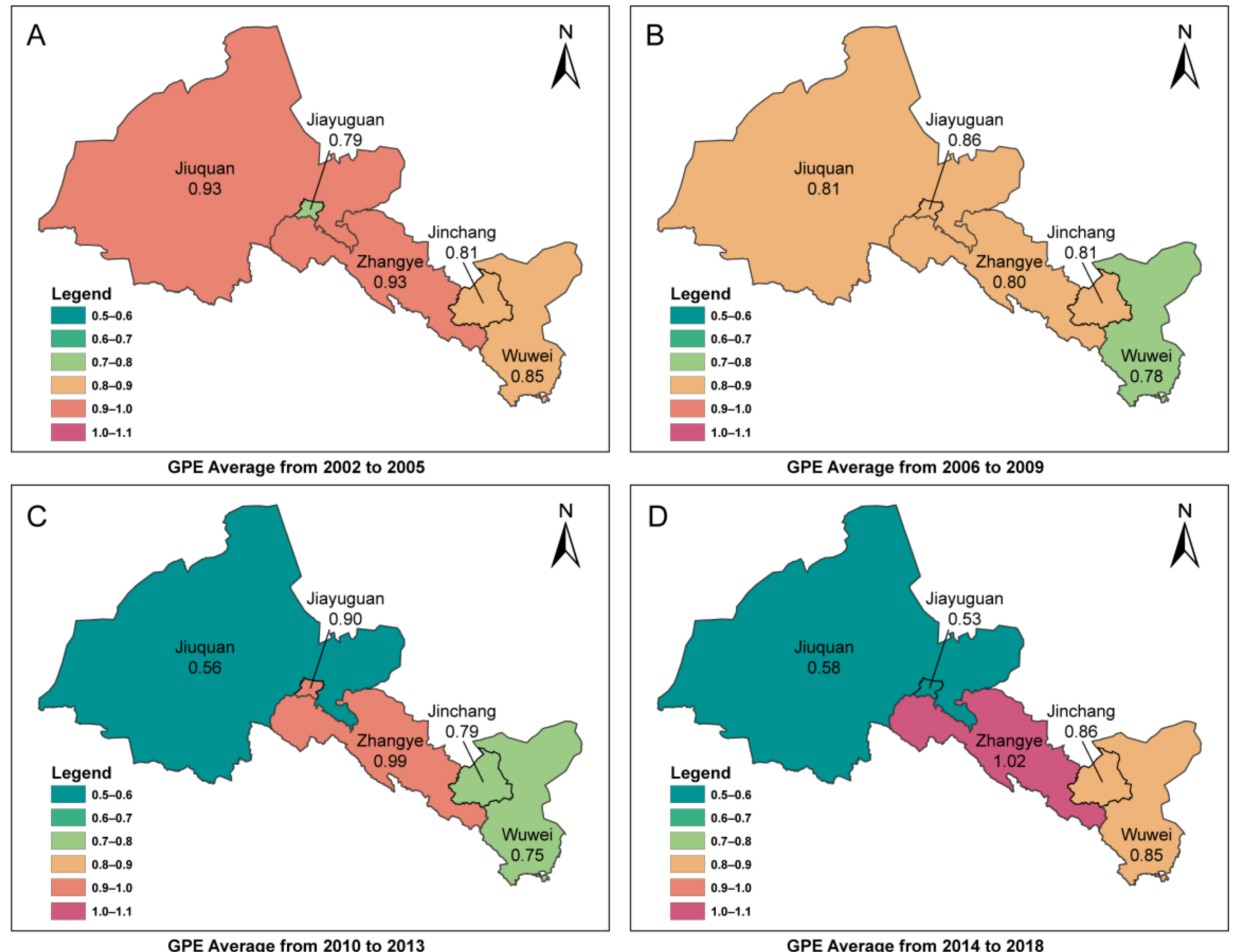

**Figure 5.** The spatial distribution of grain production efficiency in the Hexi Corridor in (**A**) 2002–2005, (**B**) 2006–2009, (**C**) 2010–2013, and (**D**) 2014–2018.

It can be seen from Figure 5 that the spatial variation in grain production efficiency in different divisions was very significant. The contrast between Jiuquan and Zhangye was very obvious. The GPE of Jiuquan area showed a decreasing trend during 2002 to 2018. The average GPE of Jiuquan division is 0.93, 0.81, 0.56, and 0.58 in corresponding four-time sections of 2002–2005, 2006–2009, 2010–2013 and 2014–2018. However, the GPE of Zhangye area showed an increasing trend during 2002 to 2018. The average GPE of Zhangye area is 0.93, 0.80, 0.99, and 1.02 in corresponding four-time sections of 2002–2005, 2006–2009, 2010–2013 and 2014–2018. Figure 5 shows that from 2002 to 2009, the GPEs of Zhangye and Jiuquan were almost the same. Since 2010, the GPE of Jiuquan has started to decline, while the GPE of Zhangye has increased steadily, which has gradually widened the gap between the two areas.

The GPE of Jiayuguan varied greatly in different time sections. During 2002–2013, Jiayuguan's GPE was in a steady increasing phase, while during 2014–2018, Jiayuguan's GPE dropped rapidly from 0.9 to 0.53, a drop of 41%. Relatively, from 2002 to 2018, the GPEs of Jinchang and Wuwei did not fluctuate much and remained basically stable (Figure 5). From the perspective of temporal distribution, the GPE of the five areas in the Hexi Corridor

remained at a high level from 2002 to 2009. The GPE of Jiuquan had dropped year by year since 2010, while the GPE of Jiayuguan had dropped significantly since 2014. In terms of spatial distribution, by 2018, as of the time of the analysis in this paper, the GPEs of Jiuquan and Jiayuguan were 0.58 and 0.53, respectively, far less than the other three regions (Figure 5).

### 3.5. The Importance Analysis of Affecting Grain Production Factors by Extra-Trees

The Extra-Trees machine learning algorithm can directly obtain the impacts of each human-driven or nature-driven factor on grain production. We set 70% of the dataset for training and 30% for testing, so the training dataset was 59, and the test dataset was 26 in total. Although the number was not very large, the $R^2$ came to 0.989 to testify to the qualification of the application of this method (Figure 6). The detailed results are presented in Table 3 and Figure 7. Human-driven factors dominated grain production, accounting for 93.7%, and nature-driven factors only accounted for 6.3% in total. Among them, GPA was the most central to grain production accounting for 37.42%. ALF and CFC were followed by GPA, and they were also over 10%, accounting for 27.7%, and 11.63%, respectively. After that, IWC and AP related to water resources accounted for the importance of grain production for 6.17% and 4.06%, respectively. The top five accounted for 86.97% in total in importance.

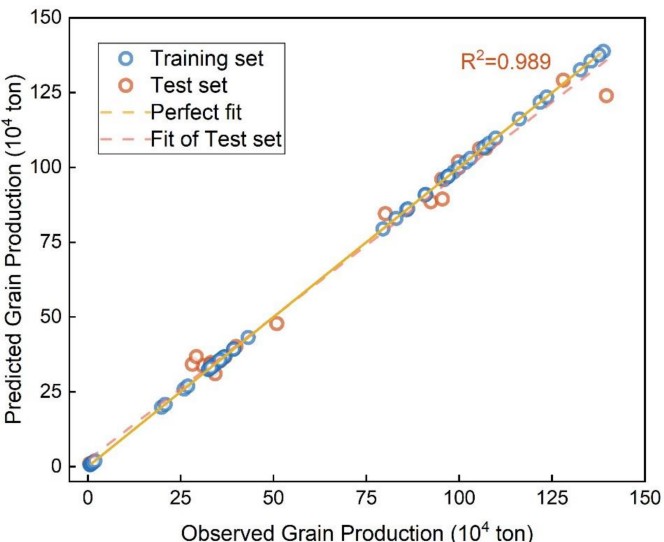

**Figure 6.** Prediction and R-squared of test set of the Extra-Trees.

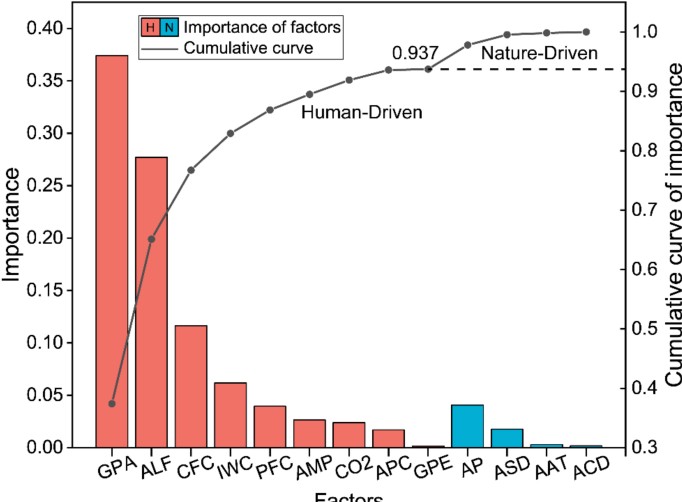

**Figure 7.** Results of the importance of factors influencing on grain production.

**Table 3.** Results of the Extra-Trees.

| Rank | Factor | Category | Importance |
|:---:|:---:|:---:|:---:|
| 1 | Grain Crops Planting Area (GPA) | Human-Driven | 37.4181% |
| 2 | Agricultural Labor Force (ALF) | Human-Driven | 27.6876% |
| 3 | Chemical Fertilizer Consumption (CFC) | Human-Driven | 11.6299% |
| 4 | Irrigation Water Consumption (IWC) | Human-Driven | 6.1719% |
| 5 | Annual Precipitation (AP) | Nature-Driven | 4.0611% |
| 6 | Agricultural Plastic Film Consumption (PFC) | Human-Driven | 3.9515% |
| 7 | Total Agricultural Machinery Power (AMP) | Human-Driven | 2.6389% |
| 8 | $CO_2$ Emissions ($CO_2$) | Human-Driven | 2.3953% |
| 9 | Average Sunshine Duration (ASD) | Nature-Driven | 1.7584% |
| 10 | Agricultural Pesticide Consumption (APC) | Human-Driven | 1.6925% |
| 11 | Annual Average Temperature (AAT) | Nature-Driven | 0.2871% |
| 12 | Area Covered by Natural Disaster (ACD) | Nature-Driven | 0.1657% |
| 13 | Grain Production Efficiency (GPE) | Human-Driven | 0.1421% |

## 4. Discussion

### 4.1. Analysis of the Variation of Grain Production Efficiency in the Research Period

The GDP of Gansu Province increased by more than five times during the study period, and its grain production increased by about 47% (NBSC, 2003–2019). However, our research shows that the average efficiency scores obtained through stage 1 evaluation of DEA did not increase during the study period (Figure 3). After removing external environmental and stochastic errors factors in stage 2, the efficiency scores were lower and even had a slight downward trend. Similar to the results of Zheng et al. [47], agricultural production efficiency had indeed decreased in the past two decades rather than rising. This means that the substantial increase in grain production was mainly due to higher inputs. In 21st century China, where science and technology have been developing rapidly, it is taken for granted that the increased grain production was mainly due to the significant improvement in technical efficiency. However, the results of our three-stage super-SBM DEA assessment are contrary to what we supposed. The proportion of input slacks had not decreased, and the efficiency had not increased. The rocketing increase in grain production was mainly due to the large-scale increase in input rather than the improvement of efficiency scores.

From the coefficients of SFA regression results in stage 2 (Table 2) and the variation between stages 1 and 3 (Figures 3 and 4), external environmental factors greatly impacted efficiency scores. From 2002 to 2018, among the scores of stage 1 to stage 3, the GPE of Jiayuguan decreased by about 0.26, followed by Jinchang 0.23, and Wuwei and Zhangye also decreased slightly, but it was relatively not obvious; in Hexi Corridor, only Jiuquan increased by about 0.17. It seems that the efficiency scores of stage 1 shrank to the compacted middle from both sides to form the efficiency scores of stage 3, making the original high-efficiency lower and the low-efficiency higher.

Most coefficients for EV1 are positive, so it means that the higher the GRP, the higher the slacks, and the lower the GPE. Jiayuguan has the highest GRP throughout our research period and a lot more than the other four cities. Therefore, after the deduction of the impacts of environmental factors, the efficiency score of Jiayuguan has a considerable drop from stage 1 to stage 3 (Figures 3 and 4). For EV2, the proportion of primary industry in GRP, most coefficients were negative, which means that a higher proportion could increase the efficiency score. The proportion of Zhangye and Wuwei was slightly higher than that of the other three cities. Hence, after all DMUs were treated in the same environment in stage 2, the efficiency scores of Zhangye and Wuwei in stage 3 were slightly lower than that in stage 1. Regarding EV3, the proportion of inner expenditures of S&T activity in GRP, similarly, most coefficients were negative, which means that a higher proportion will increase the efficiency scores. The proportion of Jiayuguan and Jinchang was much higher than that of the other three cities, so the efficiency scores of Jiayuguan and Jinchang in stage 3 were significantly lower than that in stage 1 after the process of stage 2. However, as for Jiuquan city with a poorer environment, the efficiency scores in stage 3 were higher

than those in stage 1 because we put its environment under the same scenario as in the other four cities. Jinchang and Jiayuguan have the higher value for EV4, Total Value of Imports and Exports Commodities. Only two factors, ALF and IWC, pass the 1% significant test with a positive value. Therefore, there is a certain degree of decline in efficiency scores for Jinchang and Jiayuguan in stage 3 (Figure 4).

The apparent variation of efficiency scores in stage 1 and stage 3 also confirmed the necessity of dealing with external environmental factors and stochastic errors in stage 2. It also proved that even under the strict non-radial model, the efficiency scores of DMUs would be overestimated because the influence of removing environmental factors was not considered [65–68]. To note that, although these correlations were significant, the coefficients were very close to zero. However, that does not mean EV1 and EV4 had no great influence on the input slacks and the efficiency. The magnitude of EV1 and EV4 are higher than EV2 and EV3. As a result, the coefficients may seem tiny in EV1 and EV4. This research is not aiming to check which environmental factors impact more on the slacks, so we did not perform the normalization or standardization for environmental factors. We can check the relationship by the positive or negative of the coefficients to figure out whether it makes sense.

### 4.2. Grain Production and Its Influencing Factors

By reviewing the existing studies on factors affecting grain production, we can find that the results are different due to different study areas or research methods. Zhang et al. [69] applied the Cobb-Douglas production function model to analyze the relationship between maize yield and its influencing factors in Daqing City. Among the four influencing factors, pesticide application rate had the most significant impact on maize yield, followed by planting area, the amount of fertilizer applied, and finally, the amount of available precipitation. Lu et al. [50] believe that climate change significantly impacts rice production. Pan et al. [52] fitted the relationship between grain production and potential driving factors in China through OLS regression and spatial regression. The results showed that arable land area and amount of labor are the two most important factors affecting grain production.

In our research, the Extra-Trees algorithm returned the results of importance for 2002–2018 (Table 3 and Figure 7), in which $R^2$ reached 0.989 (Figure 6). It considers Grain Crops Planting Area (37.42%) as the most important, followed by Agricultural Labor Force (27.69%), Chemical Fertilizer Consumption (11.63%), and Water Resources (10.23%) if we combined Irrigation Water Consumption (6.17%) and Annual Precipitation (4.06%). Our result is similar to the result from Pan et al. [52], in which the planting area and labor force were the most two important factors. It is interesting that the GPE was the least important. However, it was also apparent that when the grain production increased a lot throughout our study period, the GPE fell instead. Although the relationship between grain production and the other factors were probably non-linear, the GPE almost was almost ignored by our prediction model. On the other hand, we state that the new scientific techniques or equipment applied have not improved a lot in agriculture during our research period from two aspects, the decreasing, and the least importance of GPE. Even though the grain production of Hexi Corridor expanded in magnitude, the most contributors were planting area, labor force, and fertilizers rather than the efficiency from 2002 to 2018. It means that most of the increase in grain production is due to the higher input of production materials, but not the better managerial methodologies or cutting-edge technology.

In addition, Figure 7 indicated that Human-Driven factors accounted for 93.7% and dominated in the model. Only two showing the importance of nature-driven factors were over 1%, Annual Precipitation and Average Sunshine Duration. Water resources of course are important for grain production, but it is a less frequently random event in relative arid areas and cannot be comparable to irrigation water in magnitude. Additionally, Sunshine Duration is crucial to the growth of all plants. That it did not rank in a high level may be due to the time scale in our research. The yearly Average Sunshine Duration has small variance

that leads to the unimportance of the model. All these results show that Human-Driven factors were controlling the grain production rather than Nature-Driven. As a result, we cannot find too many impacts on grain production from climate change.

*4.3. Policy Recommendations*

Our research shows that from 2002 to 2018, in the absence of major natural disasters, the increase in grain production in Hexi Corridor was mainly achieved by increasing planting area, increasing labor input, increasing chemical fertilizers and irrigation, and so on. Among the various factors of grain production, the Grain Crops Planting Area is the most important factor, accounting for 37.42%. It is followed by Agricultural Labor Force, accounting for 27.69%, Chemical Fertilizer Consumption accounted for 11.63%, and Water Resources accounted for 10.23%. However, the Grain Production Efficiency (GPE) only accounted for 0.14%.

That is to say, at present, increasing the planting acreage is much more important than increasing the per unit area yield. Therefore, first, we need to raise the planting area to increase grain production. The second step is to increase labor force input, which can be achieved by increasing the proportion of agricultural mechanization—followed by other cultivation factors such as fertilization and irrigation.

Demand for food and livestock feed is rapidly increasing due to population growth, urbanization, and changing diets that include more animal-based products. One consequence of these changes has been expanding agricultural land allocated to livestock through direct and indirect use of farmland for animal feed production [70,71]. At a time when water scarcity and land degradation threaten food security, growing interest in biofuels, feed, and fiber in recent years has created conflicting demands on how agricultural land can be used.

In this regard, we put forward two suggestions: First, improve the comprehensive grain production capacity. All localities are required to strictly abide by the cultivated land red line and the permanent basic farmland control line, strengthen the construction of high-standard farmland, accelerate the construction of farmland water conservancy, actively promote the comprehensive reform of agricultural water prices, and vigorously promote the application of improved water conservancy irrigation, promotion of water-saving irrigation and other key agricultural technologies. According to the actual development, scientifically adjust the rural industrial structure and increase investment in small and medium-sized infrastructure construction.

The second is to keep the planting area and grain production basically stable. It is emphasized that policies must be stable in scope and output. The prominent grain-producing areas should give full play to their advantages. The main sales areas and the production and sales balance areas should strengthen grain production to ensure that the planting area and grain production are stable over the year. At the same time, it is necessary to protect arable land through specific and feasible protection mechanisms. In recent years, with the acceleration of urbanization, construction land and industrial land have increased, and the area of cultivated land has decreased sharply in China. Therefore, the government should limit cultivated land transfer, establish a paid protection mechanism for cultivated land, and use economic means to protect cultivated land.

In addition, our research shows that the most important impact factors of grain production in the Hexi Corridor from 2002 to 2018 were human-driven rather than nature-driven factors. At present, farmers' enthusiasm for grain production is not high, which may become a critical restrictive factor for grain production in China in the near future. The Chinese government should protect arable land and promote large-scale production. The government should pay more attention to agricultural subsidies for underdeveloped areas to increase farmers' enthusiasm for food production. At this stage, the rural agricultural infrastructure in mountainous areas is quite different from that in plain areas, and the government should increase the construction of agricultural infrastructure in mountainous areas.

Improving agricultural production efficiency is the basic requirement of developing modern agriculture. In this paper, we also found that between 2002 and 2018, the grain production efficiency (GPE) accounted for only 0.14% of grain production in the Hexi Corridor. It shows that there is still a massive space for improvement in production efficiency in the Hexi region.

From our study, it can be seen that the spatial variation in grain production efficiency in different divisions is very significant. Although the GPE of the five areas in the Hexi Corridor remained at a high level from 2002 to 2009, by 2018, the GPEs of Jiuquan and Jiayuguan were far less than the other three regions (Figure 5).

In this regard, countermeasures should be taken to improve the efficiency of agricultural production and improve the current situation of unbalanced regional development. Especially in Jiuquan and Jiayuguan areas, we need to establish technology-oriented agriculture and improve the organization of agricultural production.

The development of informatization has a promoting effect on agricultural productivity. This effect mainly comes from the improvement of agricultural technical efficiency. To make its agricultural efficiency leap forward in China, we should have a consensus on a long-term science and technology strategy. From now on, it is necessary to make essential work and strategic arrangements for the construction of agricultural informatization in the first half of this century:

The first is to build an agricultural informatization strategy, put forward a long-term plan for information agriculture construction in China, and incorporate major agricultural informatization technologies into regional development plans.

The second is to focus on strengthening the construction of agricultural science and technology, education, and economic information network centers and gradually promote and support the construction of provincial, county, and township information networks in China.

The third is to promote the construction of smart agriculture systematically. Implement the innovative agricultural big data project, establish an agricultural data governance system that is compatible with the development of big data, strengthen the development and intelligent transformation of field management application systems, and improve the level of supervision, monitoring, and management.

Informatization is the primary project for building the emergency management system in the new era. From agricultural modernization to agricultural informatization, it is an inevitable rule of agricultural development. In accelerating the process of agricultural modernization, it is necessary to accelerate the process of agricultural informatization and actively meet the challenges of the emerging new technology revolution and knowledge innovation.

## 5. Conclusions

This study assessed grain production efficiency and grain production influencing factors in Hexi Corridor, Gansu Province, China. The three-stage super-SBM DEA model was developed for efficiency score calculations and Extra-Trees machine learning algorithm was applied to determine the importance of grain production for each factor. Despite an enormous increase in grain production, the average grain production efficiency had not been improved rather than dropped from 2002 to 2018. That means, during this period, the increase in grain production may not be based on, for example, the improvement of management and cutting-edge technology inputs. Then, we introduced climate factors, Nature-Driven, combined with Human-Driven factors to build a machine-learning model by Extra-Trees algorithm. Our result shows that Grain Crops Planting Area (37.42%) was the most important factor, followed by Agricultural Labor Force (27.69%), Chemical Fertilizer Consumption (11.63%), and Water Resources (10.23%). However, the Grain Production Efficiency (GPE) only accounted for 0.14%. Furthermore, Human-Driven and Nature-Driven factors accounted for 93.7% and 6.3%, respectively. It confirmed that GPE was not a major influencing factor for the increase in grain production in Hexi Corridor from

2002 to 2018. It mostly was because of the higher Human-Driven input rather than the change of climate factors.

**Author Contributions:** Conceptualization, Z.Y. and W.Z.; methodology, Z.Y.; software, Z.Y.; validation, Z.Y., Y.W. and X.C.; formal analysis, Z.Y.; investigation, Z.Y., Y.W. and X.C.; resources, Y.W. and X.C.; data curation, Z.Y., Y.W. and X.C.; writing-original draft preparation, Z.Y. and W.Z.; writing-review and editing, Z.Y. and W.Z.; visualization, Z.Y.; supervision, W.Z.; project administration, W.Z.; funding acquisition, W.Z. All authors have read and agreed to the published version of the manuscript.

**Funding:** This research received no external funding.

**Institutional Review Board Statement:** Not applicable.

**Informed Consent Statement:** Not applicable.

**Data Availability Statement:** All the data used in this study is public.

**Acknowledgments:** We gratefully acknowledge Robert Taylor and Stephen Leong for their generous help and support in funding resources. We thank Gansu Statistics Bureau, Gansu Hydrology, and Water Resources Survey Bureau for providing data in our research. We also would like to thank the reviewers and editors for their thoughtful comments and efforts towards improving our manuscript.

**Conflicts of Interest:** The authors declare that they have no known competing financial interests or personal relationships that could have appeared to influence the work reported in this paper.

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
