# Peer review of "Comprehensive Analysis of Grain Production Based on Three-Stage Super-SBM DEA and Machine Learning in Hexi Corridor, China"

_sustainability, doi:10.3390/su14148881_

Round 1

Reviewer 1 Report

Good Work. Improve the clarity of the presentation.

Author Response

We thank the referee for the careful and insightful review of our manuscript. We address all the concerns of the referee here.

Response to Reviewer 1

Thanks for the formatting suggestion. We have changed the reference format in both main manuscript and citation part to follow the MDPI format. Please find all the changes in the trackable word version.

The following are the comments on this paper:

  1. Comment 1. Abstract: Although written well, it looks lengthy. Revise the abstract section by highlighting the purpose, methods used, outcome, and implications of the study. Keep the abstract within 250 words.

Response: Thanks for this suggestion. We have revised the Abstract and lower down the size from 308 words to 190 words with clear purpose, methods, outcome, and the implications.

  1. Comment 2. Introduction: Written well. But looks lengthy. Try to revise the introduction section by clearly stating the need and significance of the study, findings of the existing studies, what misses in the existing literature, list the research gaps, contribution of the study. Avoid redundant data in the introduction. Keep the introduction crisp and clear.

Response: Thank for this good suggestion and instructions. The introduction has been revised and some redundant and unnecessary citations were removed. And some new citations are added. We make the introduction to be more concise and downsize it from total 2602 words to 1834 words. All the changes are trackable in word version.

  1. Comment 3. Thumb rule of the DEA method needs to be discussed. Methodology needs to be explained better. Three-stage super-SBM DEA needs better explanation.

Response: Thanks for this wise suggestion that we ignored when doing the manuscript. We added more reasonable basic concepts and the formula of the traditional DEA method and three-stage super-SBM DEA in the Methodology section to make the whole section smoother and more precise. Additionally, we change the rank of the formula number due to the addition of the traditional DEA method.

  1. Comment 4. Discussions need to be improved.

Response: Thanks for this suggestion. We have double checked the discussion and made the necessary changes. For example, the original lines 495-498 are deleted; lines 552-554 are deleted; lines 557-564 are deleted. Some ideas are abounded, such as original lines 599-600, lines 634-35, lines 665-666; lines 668-671 are deleted. Pleas find the trackable changes in word version. 

  1. Comment 5. References: Not in the journal format. Use Mendeley software. Better refer recent articles rather than old articles.

Response: Thank you for this suggestion. The references are revised as MDPI format in both of main manuscript and citation part. The changes are trackable in word version.

  1. Comment 6. Provide the tables and figures in the main manuscript. Table and figures must be placed immediately after the first mentioning.

Response: Many thanks for this suggestion. All table and figures are inserted in the main manuscript where they are first mentioned.

Reviewer 2 Report

This is an interesting subject. The topic of grains has become more and more important, especially in the current context.

However, this paper has some shortcomings and needs improvement.

The abstract is too long and should be reworded to better summarize the content of the article. According to the journal's recommendations, a maximum of 200 words is accepted.

At the beginning of the article, reference is made to a paper from 2003 to present the situation in recent years, especially since there is talk of funding for agriculture and investment. Such an assessment should be updated with much newer data.

The source for the figures representing maps should be specified. The Acknowledgment section is also unclearly worded.

Although the English style is generally acceptable, there are certain parts where the text should be revised. I also noticed the repetition of words in the same sentence.

Given the opening of the journal to Free Format Submission, I assume that the adaptation of the format to the suggested template will be done at a later stage.

Author Response

We thank the referee for the careful and insightful review of our manuscript. We address all the concerns of the referee here.

Response to Reviewer 2

This is an interesting subject. The topic of grains has become more and more important, especially in the current context.

However, this paper has some shortcomings and needs improvement.

  1. Comment The abstract is too long and should be reworded to better summarize the content of the article. According to the journal's recommendations, a maximum of 200 words is accepted.

Response: Thank you for this suggestion. We have revised the abstract to be more compact and concise to 190 words in total.

  1. Comment 2. At the beginning of the article, reference is made to a paper from 2003 to present the situation in recent years, especially since there is talk of funding for agriculture and investment. Such an assessment should be updated with much newer data.

Response: Thank you for your suggestion. The references are replaced by the updated newest publications in the years 2021, 2021, and 2018, respectively. Please find the changes in the trackable word version.

  1. Comment 3. The source for the figures representing maps should be specified. The Acknowledgment section is also unclearly worded.

Response: Thanks for this suggestion, we have made the necessary changes and added the source of maps in section 2.1, which are from the Standard Map Service System and the Resource and Environment Science and Data Center. The acknowledgment is revised: “Thanks to Gansu Statistics Bureau, Gansu Hydrology, and Water Resources Survey Bureau for providing data in our research. We would like to thank the reviewers for their thoughtful comments and efforts towards improving our manuscript.”

  1. Comment 4. Although the English style is generally acceptable, there are certain parts where the text should be revised. I also noticed the repetition of words in the same sentence.

Response: Thank you for this question. We have carefully checked the spelling and grammar in the manuscript and corrected the typos and errors we can find. Please find all the changes in the trackable word version.

  1. Comment 5. Given the opening of the journal to Free Format Submission, I assume that the adaptation of the format to the suggested template will be done at a later stage.

Response: Thanks for this suggestion. We have changed the reference format in both the main manuscript and citation part to follow the MDPI format. Please find all the changes in the trackable word version.
